# Development of an intervention to facilitate dissemination of community-based training to respond to out-of-hospital cardiac arrest: FirstCPR

**Sonali Munot** [1]*, **Janet Bray**[2], **Adrian Bauman**[3], **Emily J. Rugel**[1], **Leticia Bezerra Giordan**[1], **Simone Marschner**[1], **Clara K. Chow**[1,4], **Julie Redfern**[4,5]

1 Faculty of Medicine and Health, Westmead Applied Research Centre, The University of Sydney, Sydney, Australia, 2 Department of Epidemiology and Preventive Medicine, Monash University, Melbourne, Victoria, Australia, 3 Faculty of Medicine and Health, School of Public Health, The University of Sydney, Sydney, Australia, 4 The George Institute for Global Health, University of New South Wales, Newtown, Australia, 5 Faculty of Medicine and Health, School of Health Sciences, The University of Sydney, Sydney, Australia

* sonali.munot@sydney.edu.au

**Data Availability Statement:** Data cannot be shared publicly because the materials and resources that were reviewed for the purpose of

## Abstract

### Background and aim

Out-of-hospital cardiac arrest (OHCA) is a significant public health issue with low survival rates. Prompt bystander action can more than double survival odds. OHCA response training is primarily pursued due to work-related mandates, with few programs targeting communities with lower training levels. The aim of this research was to describe the development process of a targeted multicomponent intervention package designed to enhance confidence and training among laypeople in responding to an OHCA.

### Methods

An iterative, three-phase program development process was employed using a mixed methods approach. The initial phase involved establishment of a multidisciplinary panel that informed decisions on key messages, program content, format, and delivery modes. These decisions were based on scientific evidence and guided by behavioural theories. The second phase comprised the development of the intervention package, identifying existing information and developing new material to fill identified gaps. The third phase involved refining and finalising the material via feedback from panel members, stakeholders, and community members.

### Results

Through this approach, we collaboratively developed a comprehensive evidence-based education and training package consisting of a digital intervention supplemented with free access to in-person education and training. The package was designed to teach community members the specific steps in recognising and responding to a cardiac arrest, while addressing commonly known barriers and fears related to bystander response. The tailored

this study, are owned by third parties thus limiting authors' sharing capabilities.

**Funding:** Funding was obtained from the National Health and Medical Research Council (NHMRC), Australia [partnership project grant, number APP1168950]. In addition, monetary support was received from New South Wales Health and The University of Sydney Deputy Vice-Chancellor (Research) [COVID-19 Support Scheme Grant record ID: G201443]. All partner organisations provided in-kind support towards the project [see organisation list in Supplementary section S1]. The funders had no role in study design, data collection and analysis, decision to publish, or preparation of the manuscript. JR is supported by a NHMRC Investigator Grant [GNT2007946], CC is supported by a NHMRC Investigator Grant [APP1195326], JB is supported by a Heart Foundation Fellowship [#104751] and SMunot is supported by a PhD scholarship from the Westmead Applied Research Centre.

**Competing interests:** The authors have declared that no competing interests exist.

program and delivery format addressed the needs of individuals of diverse ages, cultural backgrounds, and varied training needs and preferences.

## Conclusion

The study highlights the importance of community engagement in intervention development and demonstrates the need of evidence-based and collaborative approaches in creating a comprehensive, localised, relatively low-cost intervention package to improve bystander response to OHCA.

## Introduction

Out-of-hospital cardiac arrest (OHCA) is a challenging public health problem resulting in significant mortality and morbidity globally [1]. Fewer than one in ten individuals survive to hospital discharge following an OHCA, and wide variation is reported across jurisdictions in arrest characteristics and outcomes [2]. Receipt of basic life support including cardiopulmonary resuscitation (CPR) and defibrillation from bystanders more than doubles survival [3]. Trained bystanders are more likely to be confident to respond [4], and engaging community members to receive training is an important step to create a 'culture of action' [5, 6].

Internationally, implementation of CPR education is variable. CPR training is mandated in some jurisdictions, such as parts of North America and Denmark [7, 8]. Surveys of the public show the most common way community members receive CPR training is though employment, such as when it is mandated for those working in sectors such as health, community services, fitness, security, and public transport [9]. For many community members, CPR training is not a requirement, and common barriers to voluntary CPR training include cost, lack of time, lack of motivation, and lack of awareness [4, 10]. Community CPR education and training initiatives can improve bystander response [11], as indicated by the fact that regions with high CPR training rates have higher rates of bystander CPR [12]. Conversely, passive approaches to diffusion of research information are often ineffective and do not occur spontaneously and naturally [13]. A survey of over 250 public health researchers identified that comprehensive, multi-level approaches, guided by theory and targeted to specific audiences are likely to be most effective [14]. Despite these broad findings, there is little specific knowledge on how CPR training and education on response to OHCA can be disseminated to community members efficiently [11].

The aim of this research is to describe the process and iterative steps taken in the development of a tailored public education and training program with a design approach that gives consideration to locally relevant culture and context.

## Materials and methods

Our formative evaluation comprised a three-phase mixed-methods process to develop an intervention package aimed at increasing the public's knowledge and awareness of OHCA as well as their skills and confidence in performing CPR and using a defibrillator. The program was designed to be implemented at low-cost through targeted promotion to sports and social community organisations and workplaces [15]. Ethics approval was obtained from The University of Sydney Human Research Ethics Committee (2020/537).

## Phase 1: Initial planning

**Stakeholder partnership and consultation.** Community-based interventions are strengthened by input from different stakeholders who can offer scientific expertise as well as practical experience [16]. A working group was formed comprising 15 members, including professional stakeholders with knowledge and experience in clinical care, public health, epidemiology, and research, along with governmental and non-governmental organisation (NGO) stakeholders with knowledge of and practical experience with community training needs and local preferences (S1 File: List of stakeholder organisations).

One in-person meeting, and three virtual/online meetings were convened to reach consensus on program components through a discussion of the current literature, evidence, and guidelines. These sessions informed the intervention's overall format, content, key messages, and delivery plan (mode, frequency, dose, duration). Important considerations included: customisation and tailoring for target groups; considering needs of older adults, and culturally and linguistically diverse community members; and issues of digital and health literacy. In addition, feasibility, acceptability by the intended training population, and addressing known barriers and enablers to uptake were considered.

**Development of material guided by evidence, guidelines, and behavioural theories.** A review of the published literature, current practices, and guidelines in CPR education and training informed the collation and development of intervention material [17]. The primary intent of the intervention package was: to a) to impart education and provide access to training on responding to OHCA; and b) positively influence behavioural intention or 'willingness to act'. The intervention's design and development were informed by The Theory of Planned Behaviour, Theory of Reasoned Action, and the COM-B framework, as well as by theories that discuss elements needed for community-level change [18, 19]. The COM-B framework (Fig 1) describes key elements for behaviour change as capability (C), opportunity (O), and motivation (M) to perform a behaviour (B). Capability in this context can be seen as feeling psychologically confident and physically capable to perform CPR or use an Automated External Defibrillator (AED), and motivation in this study is likely to be reflective (i.e., developing the intention to act). When developing the intervention components, the initial focus was specifically on developing elements that addressed the aspects of developing the 'capability' (psychological) and 'motivation' (reflective) needed when the 'opportunity' arises for a bystander to respond.

Other than individual determinants of behaviour change, broader factors such as culture, geography, social structure, and socioeconomic status can play a role in influencing change at the community level. Public health interventions are likely to be more effective when this ecological perspective is considered and when they are designed to reflect the social milieu and broader community characteristics of the target audience [20]. The stakeholder group also drew on social marketing approaches and diffusion of innovation theory to inform specific intervention components [19].

## Phase 2: Development of intervention package

**For digital delivery.** For a comprehensive review of the existing digital material, an internet search identified educational, informative, and motivational content in the form of videos and factsheets. Numerous publicly available videos and websites were reviewed and considered. We aimed for the program development process to be replicable by groups in other settings and with limited resources, so applying a process to identify well-designed existing resources was important [21]. Two members of the research team (SM and LBG) independently searched for material related to 'responding to a cardiac arrest' from the sources above,

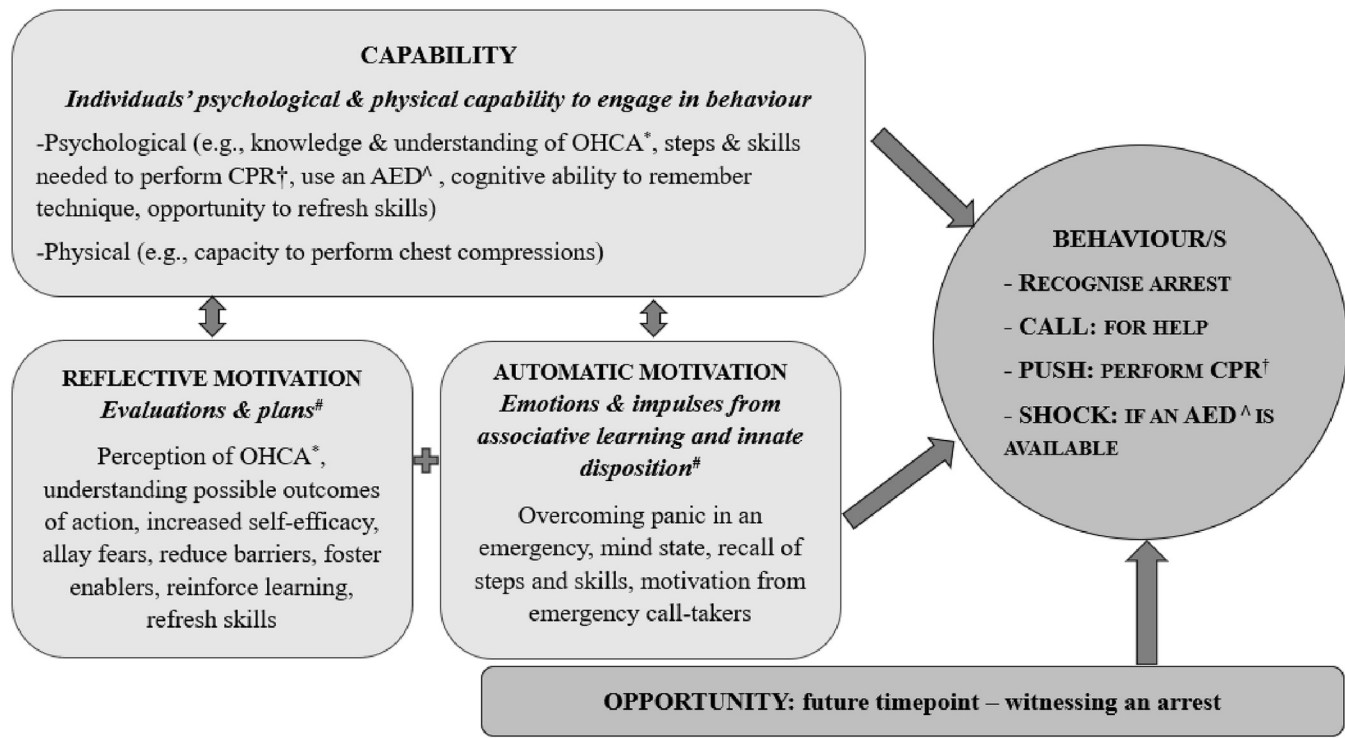

*Out-of-hospital cardiac arrest; †Cardiopulmonary resuscitation; ^Automated external defibrillator; #Definitions adapted from Michie et al 2011

**Fig 1. Theory-guided approach informing intervention development (image concept and flow adapted from COM-B model) [18].**

as well as on government, ambulance services, and relevant NGO websites. These materials were summarised in a spreadsheet using pre-defined screening criteria (see S2 File).

Aligned with the key messages for the campaign identified in phase 1, the research team created a bank of direct and succinct sentences to create interest and serve as a hook to persuade participants to view the videos and factsheets. Previous studies describing the process of development of a bank of text messages designed to support health behaviour change guided our approach [22]. The overarching aim among team members was to keep these sentences simple and short, be informative and/or motivational, and feel personalised (e.g., greeting message recipients by their name).

**For face-to-face/in-person delivery.** Accredited training involves in-depth and comprehensive activities and necessitates practicing CPR skills on a mannikin. Such training in Australia typically incurs a cost, making it less accessible to many community members. To overcome this obstacle, providing vouchers to attend accredited training for free was agreed to by all partners and approved by the steering committee. Such 'incentivisation' in driving change for discrete and time-defined behaviours (including attending educational and training sessions) has been reported as effective in reviews of behaviour change studies [23]. These sessions were designed to be delivered over two to three hours and to smaller groups at participating community organisations. To complement this formal training, a shorter, informal education session including a brief demonstration of CPR and use of an AED would also be made accessible to larger groups. These sessions were envisaged as an interactive introduction to the main concepts, addressing barriers, and encouraging uptake of formal training to reinforce learning and practice of skills. Research team members collaborated with certified CPR

trainers to develop visual aids (slide presentation on Microsoft PowerPoint™) to present at these sessions, that were then reviewed and approved by the lead investigators.

## Phase 3: Program review and finalisation

Feedback was solicited on a total of 18 shortlisted messaging sentences and accompanying identified material (videos and factsheets). A link was sent via e-mail and consent was *implied* if participants proceeded to click the link to provide anonymous feedback or they could choose to opt out by not proceeding to click the link. Survey questions were drafted by the study team and kept simple and short given the time burden of reviewing a list of items.

Respondents were asked to provide basic demographic information, rate whether they agreed/disagreed (5-point Likert scale) if the message sentence was easy to understand, and asked whether the information provided was useful. Finally, an open-ended question asked them to make comments and suggestions about the message and accompanying material (video/factsheet) they reviewed. Each participant was asked to review a set of nine items each to avoid survey fatigue. Participants included investigators, key stakeholders, work colleagues, family, and friends. Following this process, an internal meeting reviewed the feedback and determined the final set of existing materials to be incorporated in the program, as well as drafting of new materials to fill identified gaps.

Visual resources prepared for the informal educational sessions were reviewed by core study team members to ensure all key messages were conveyed. The accredited training sessions follow the national curriculum; hence revision was not considered, and this training will be delivered as per the approved current curriculum.

## Results

Fig 2 gives an overview of outputs developed during the three-phase project.

## Phase 1: Initial planning outcome

Stakeholder groups with a shared purpose of improving outcomes for OHCA included organisations involved in CPR training, community organisations, government organisations and research organisations. A literature review and stakeholder consultation led to consensus on the intervention's key messages, all of which were evidence-based and followed current guidelines [24–27]. Content that addressed known barriers in performing CPR or using an AED was incorporated in the messaging to address known fears and barriers to providing CPR [28, 29]. In addition, survivor and bystander rescuer stories were used to inspire and encourage motivation to learn and to act. (Fig 3: Key messages box). An important consideration was to design the education package such that it was tailored to resonate with the target audience. The educational material would be customised for delivery in English and three more commonly spoken languages in the study area and bilingual interpreters would be made available for in-person sessions where necessary.

The decision to deliver educational content digitally was based on the availability and accessibility of technology, the feasibility of wide-scale implementation, and utilising appropriate technology. These issues were explored in discussions with stakeholders [30, 31]. Although digital education was selected to increase the intervention's reach and for scalability, these discussions acknowledged that learning in traditional ways may be preferred by some community members [4, 32]. The digital component will be delivered using text messages, e-mail, and social media, supplemented by access to in-person education and accredited training. This approach is flexible, enabling outreach to various segments of the population, depending on their specific preferences for training.

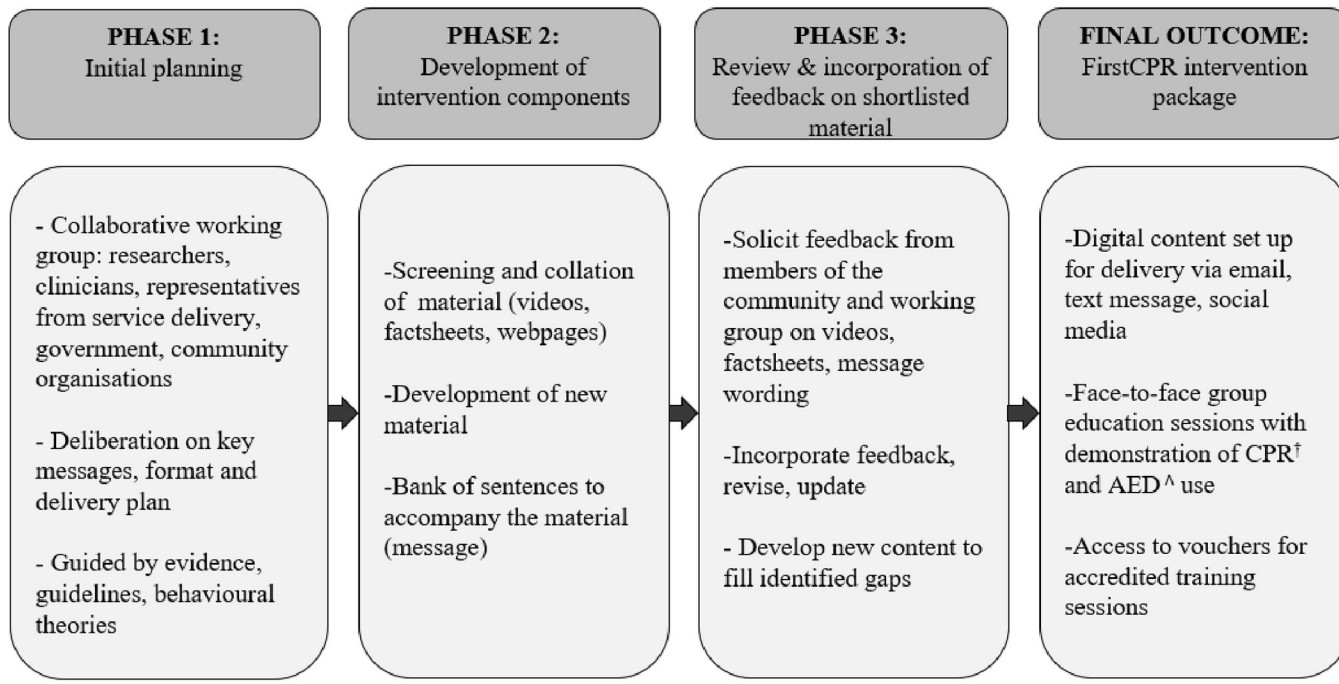

†*Cardiopulmonary resuscitation;* ^*Automated external defibrillator*

**Fig 2. Summary of intervention development phases.**

## Phase 2: Outcome of screening and development of intervention components

Initial electronic searches identified a substantial number of existing materials, with approximately 100 publicly available videos identified. These findings were supplemented by recommendations from partners, with all materials screened in detail and 11 videos selected. Topics included demonstrations of what do in the event of a cardiac arrest with a focus on compression-only CPR; how to perform CPR and use a defibrillator; popular personalities noting the importance of CPR in saving lives; narration of survivor and bystander stories; and facts, myths, and commonly known barriers. In addition, approximately 20 factsheets were screened, with three selected. Four additional factsheets were created to supplement publicly available ones and to address the campaign's key messages. Factsheets discussed tips on locating public AEDs; described key steps in response to cardiac arrest; highlighted an app that could help provide emergency services with the arrest location; and addressed myths, facts, and common barriers to bystander response.

An initial bank of messages was created to accompany the delivery of these videos and factsheets to participants. These messages were posed as questions, motivational comments or simply factual statements. (e.g., "*Most cardiac arrests occur at home. CPR can double the chance of survival. Click the link to watch a brief video explaining how to respond to a cardiac arrest in the community*".) See S3 File for examples of message sentences.

## Phase 3: Program review and intervention finalisation

In total, 64 people were recruited using snowballing approaches to provide online feedback on one of two sets of nine items each. Set 1 (five videos and four factsheets plus nine accompanying messages) was reviewed by 28 people and the remaining 36 respondents reviewed Set 2

**Box (Figure 3): Key messages reached by consensus and guided by evidence-based guidelines**

1. What is out-of-hospital cardiac arrest; how to recognise it

2. How to perform CPR[†]: compression (position of hands, depth, rate) and how to use an automated external defibrillator (AED[Λ])

3. 'Call, push, shock' – (Hands-only or compression-only CPR) and provide opportunity to learn standard CPR[†] (compressions with rescue breaths) via accredited training sessions

4. Addressing commonly known barriers (e.g., fear of performing CPR[†] incorrectly, harming the victim, disease transmission, being sued)

5. Any resuscitation is better than none

6. Cannot do harm to the patient

7. Good Samaritan law

8. Anyone can do CPR[†]

9. Anyone can use an Automated External Defibrillator (AED[Λ])

10. Importance of regular refreshers (particularly among family of people with heart conditions)

*Out-of-hospital cardiac arrest; †Cardiopulmonary resuscitation; ΛAutomated external defibrillator

**Fig 3. Box: Key messages reached by consensus and guided by evidence-based guidelines.**

(six videos, three factsheets, and nine messages). About three-quarters of respondents identified as health professionals or researchers and 26% as consumers. Two-thirds were female, a third were less than 35 years old, and most (90%) were under 65 years old.

The majority (80%) of respondents agreed/strongly agreed that messages were useful. Participants suggested that some messages should be shortened and emphasised the use of lay words and phrases. Participants preferred videos that were clear with a concise and simple demonstration of the steps required to perform CPR and provide defibrillation. They appreciated videos that enacted a real-life scenario and had practical tips such as the use of a song like "Staying Alive" to indicate the pace of the compressions. Some suggested the alternative use of a more familiar song (e.g., "Jingle Bells") to reach a wider audience. Survivor and bystander stories were considered inspiring and motivational, though respondents noted these would be more useful for individuals who have already been taught about CPR. Suggestions were made to drop videos that were long, repetitive, distractive (e.g., changing narrators constantly, unnecessary images or jokes), and those with British or American accents. Videos with irrelevant local content (e.g., a non-Australian emergency number), those narrowing the target audience (e.g., reference from a TV show that only Australian older adults were likely to have watched), and those causing potential cultural clashes were also excluded. The importance of having subtitles or voiceover in the videos was noted.

Factsheets that were short with few words and "direct to the point" messages were found to be most useful. Respondents emphasised that images needed to be clear, and pages should be less cluttered.

**Table 1. Summary of feedback on videos and factsheets.**

| Videos | | |
|---|---|---|
| **Type / category of material** | **Features that received positive comments or were liked by reviewers** | **Features that received negative comments or were less liked by reviewers** |
| Demonstration videos (explanation and demonstration of steps involved in bystander response) | • Simple clear and objective explanation<br>• Complete demonstration with all steps in clear sequence | • Presented silly jokes that could undermine the seriousness of a cardiac arrest<br>• Emergency number not local (i.e., British/American)<br>• Accent was not local (i.e., British/American) |
| Videos with explanation of myths and facts related to CPR | • Addressed common concerns and fears | • Videos with jargon<br>• Phrases that could sound scary to a lay person<br>• Too much information |
| Videos that narrated survivor or bystander stories | • Emotive and compelling<br>• Short in length | • Long length |
| Provided encouragement to act in cardiac arrest/narration only without demonstration of steps | • Highlighted importance of community action before ambulance arrives<br>• Clarified that emergency call-taker will guide the responder | • Long length<br>• Too many actors narrating was distracting<br>• Encouraging, but not explaining why, makes the video vague |
| **Factsheets** | | |
| **Type / category of material** | **Features that received positive comments or were liked by reviewers** | **Features that received negative comments or were less liked by reviewers** |
| Stated facts / provided information | • Short, clear, and fewer words<br>• Good visuals<br>• Taught something new or less known | • Too many words, or too cluttered |
| Factsheet linked to a website with information on cardiac arrest | None | • Lengthy<br>• Website had technical issues |
| Factsheet addressed barriers | • Short, clear | • Used words like 'sued'(American)<br>• Suggestions on visuals |

In summary, respondents found messages and materials useful when they were simple, clear, and short; demonstrated steps of CPR or AED use in a concise manner; did not offend subgroups of the population; and used lay terminology (see Table 1). Revisions were also suggested to the introductory messages to further simplify them and encourage the use of plain English.

Following the feedback, seven videos were excluded, and three factsheets were updated and improved. An additional search was conducted to identify three new videos. In addition, two videos and four factsheets were developed by the research team in consultation with the core team. The final set of videos and factsheets included in the intervention package was simple and succinct, with clear messaging and clinically accurate details, was produced in Australia, and listed the local emergency number.

## Final multicomponent intervention program

The final intervention package (see Fig 4) comprised a digital component and face-to-face education and training sessions. The digital material was designed for personalised and direct delivery to individual participants as well as for use in social media and newsletters. The education and training sessions were designed to be delivered in a group setting and facilitated by visual aids (presentation on Microsoft PowerPoint™) and props (mannikins and training defibrillators).

For the digital component, a final set of 18 items comprising eight videos and ten factsheets were selected (see S3 File: Sample of videos and factsheets). These were then translated or

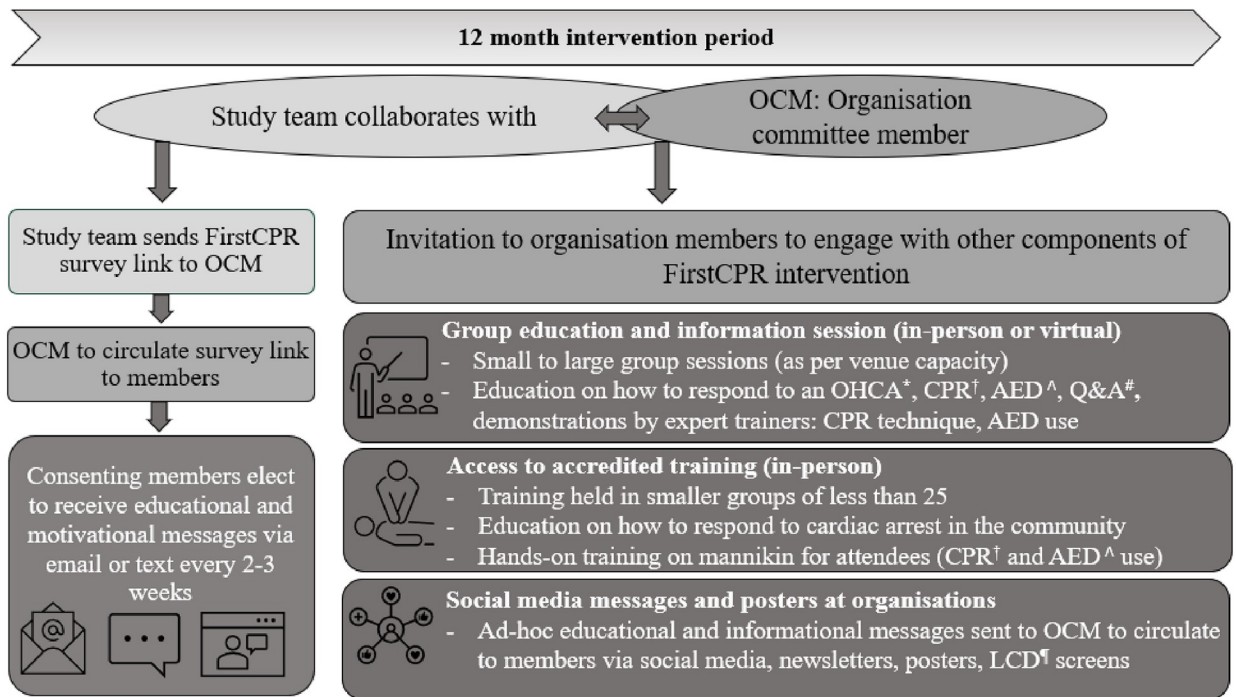

*OHCA: Out-of-hospital cardiac arrest; †CPR: Cardiopulmonary resuscitation, ^AED: Automated external defibrillator; #Q&A: Question and answer; ¶LCD screens: Liquid Crystal Display screens*

**Fig 4. Multicomponent intervention (digital and in-person format) with delivery plan to individuals and groups.**

subtitled in three other commonly spoken languages in Australia (Arabic, Chinese, Vietnamese), and set up for delivery via e-mails and text messages scheduled every two to three weeks over a one-year period. Factsheets were prepared so they could be printed for display as posters on noticeboards at participating community organisation venues.

Brief educational sessions (45–60 minutes duration) were designed for in-person group delivery, and to be conducted by experienced trainers, and with a focus on the key elements of bystander response to OHCA (what constitutes a cardiac arrest, steps in response, addressing key barriers and known fears), along with hands-on demonstrations of CPR and AED. Visual aids are to be used and the session is designed to be interactive with the opportunity for the audience to ask questions. In addition, a voucher system was set up to provide access to two-hour accredited training sessions. The research team agreed that an interpreter would be arranged for these training sessions if necessary and they would be held at venues convenient to members of each participating community organisation. The option of virtual delivery would be considered if COVID-19 restrictions impede the delivery of sessions at participating organisations during the study period. Attendance at online accredited training sessions was designed such that an in-person, hands-on training component would be held at some latter timepoint for the purpose of accreditation.

## Discussion

This paper describes the formative evaluation process that underpinned the development of an evidence-based, community-centred intervention designed to improve outcomes following OHCA that relied on a collaboration amongst volunteer-run organisations, health professionals, consumer organisation representatives, researchers, and academics. The intervention will

be implemented at the community level across urban and regional areas in New South Wales, Australia and will address multiple barriers to responding to OHCA—including access to training, lack of time, preference for digital or traditional training methods, and access to materials translated in language of choice. To aid implementation, additional funds were invested into translation of intervention materials into three commonly spoken non-English languages in the study areas, fostering access among minority groups. In addition to being locally relevant, the leveraging of partnerships between academic researchers and local community organisations resulted in the creation of a relatively low-cost program. The key ingredients of our approach included the use evidence-based material that was scientifically accurate, locally created or produced, with localised features and references that would resonate with the target audience by considering elements such as language, humour, accent, local culture, and mention of the local emergency number. Our overall approach, process, and template could be replicated in other locations with suitable and relevant adaptations to meet the needs of the target audience.

Using a co-design approach to develop community-based interventions can add significant value to public health research [33]. Although such co-design strives to align researchers' aims with end-users' needs, meaningful consumer and stakeholder engagement comes with an array of challenges related to matching researchers' and end-users' contexts, goals, evaluation metrics, expertise, and resources, as well as additional time to facilitate the intervention [34]. Despite these challenges, striving to consult where suitable and engage broadly is known to result in both higher levels of participation and improved outcomes.

The design of this intervention capitalised on the widespread use of communication technology. Although there was a focus on digital components, the importance of providing access to hands-on training opportunities cannot be overstated [35]. Alternate strategies designed to increase community awareness of OHCA and improve access to relevant education have been trialled previously, with mixed results [11]. Direct mailing out of a 10-minute videotape with CPR instructions to households was found to be ineffective in increasing bystander CPR rates [36]. A mass community-wide CPR training effort delivered the PUSH course, a 45-minute training program with an emphasis on simplified compression-only CPR, to a fifth of all residents of a medium-sized city in Japan over five years, and found improvements in the quality of CPR, but no differences in the rate of bystander CPR or survival [37]. The authors noted the possibility that the population reached was too small to impact overall bystander CPR rates; in addition, training groups mainly comprised school-children and the effort did not target older adults, who are more likely to be bystanders of an OHCA. An evaluation of bystander response before and after the implementation of the Heart Safe communities (HSC) training program in Minnesota reported a significant increase in CPR and AED use [38]. However, such HSC programs are resource-intensive, involving a comprehensive effort to increase awareness, provide training, and place public AEDs in strategic locations. Furthermore, communities in the HSC program were non-urban and had self-selected to participate [39].

Although less comprehensive than HSC, we have designed a comprehensive package that can be delivered widely at relatively low cost to various demographic groups with varying learning needs and preferences. Vetting high-quality, publicly available materials and using a systematic process to develop new materials based on stakeholder feedback is a frugal way of customising a health education campaign in limited-resource settings. Our approach also aims to increase the availability of education and training within communities that are often left out of such interventions, reducing inequities in access.

Furthermore, access to local community groups via digital means and integrated with community leadership offers an opportunity to expand confidence and CPR and AED training to the wider community. Given a dissemination strategy via community organisations, the

importance of identifying champions (whether committee members or leaders at participating organisations) cannot be overstated [40]. These leaders can build trust, facilitate logistics, and motivate individuals to participate in education and training sessions, offering a valuable conduit to the wider community in addition to their specific membership group.

As with all efforts, there were limitations in the design and development of this intervention package. First, a more in-depth community-consultation and participation process (e.g., focussed group discussions with members representative of the target audience) could have been incorporated in the design phase. We had a non-random sample (e.g., proficient in English) of participants provide feedback on the program's components and the co-design team was primarily based in Sydney, Australia, potentially limiting generalisability. However, given that key stakeholders have extensive experience working with communities and detailed practical knowledge of local preferences, their contributions provided significant insights during the initial development phase. Anonymous respondents that provided feedback on the shortlisted educational material were a mix of people from different age groups, a mix of genders, and identified themselves both as consumers and as professionals or experts.

Second, more extensive pilot-testing of the final package with a representative sample of the target audience could have identified issues related to the feasibility of delivery and engagement which, in turn, could have resulted in refinement of the intervention package to encourage engagement and improve outcomes. Unfortunately, both pilot-testing and broad community engagement were somewhat limited due to COVID-19 restrictions.

Third, incentivisation through vouchers for accredited training may not be a sustainable strategy to achieve lasting behaviour change. Instead, a more targeted strategy may be required that strikes a balance between available resources and maximum impact by delivery to populations least likely to access training based on sociodemographic features (e.g., government-funded CPR training courses for newly landed migrants).

## Conclusion

This study illustrates how collaboration amongst community members, community organisations, academics, and public health experts can effectively create a relatively low-cost, implementable intervention package to address OHCA. Looking forward, this intervention will be evaluated via a large-scale community-level cluster randomised trial, carried out in selected areas across New South Wales, Australia (the FirstCPR study [15]), to determine the intervention's effectiveness and impact on community members' self-reported training and willingness to perform CPR.

## Supporting information

**S1 File. List of stakeholder organisations.**
(DOCX)

**S2 File. Screening criteria for selection of publicly-available material (videos and fact-sheets).**
(DOCX)

**S3 File. Sample of messages and material in final digital package.**
(DOCX)

## Acknowledgments

The authors would like to thank the FirstCPR investigators and steering committee members, as well as the following people for their support and contributions towards this project: Aaisha

Ferkh, Amie Cho, Caroline Wu, Daniel Gay, Huong Ly Tong, Karina Fretwell, Marianne Gale, Mark Miller, Sophie Dyson. The authors also acknowledge the technical assistance of Cameron Fong of the Sydney Informatics Hub, a Core Research Facility of the University of Sydney.

## Author Contributions

**Conceptualization:** Sonali Munot, Janet Bray, Adrian Bauman, Simone Marschner, Clara K. Chow, Julie Redfern.

**Data curation:** Sonali Munot, Leticia Bezerra Giordan.

**Formal analysis:** Sonali Munot, Leticia Bezerra Giordan, Clara K. Chow, Julie Redfern.

**Funding acquisition:** Janet Bray, Adrian Bauman, Simone Marschner, Clara K. Chow, Julie Redfern.

**Investigation:** Sonali Munot, Janet Bray, Leticia Bezerra Giordan, Simone Marschner, Clara K. Chow, Julie Redfern.

**Methodology:** Sonali Munot, Janet Bray, Adrian Bauman, Clara K. Chow, Julie Redfern.

**Project administration:** Sonali Munot.

**Resources:** Clara K. Chow.

**Supervision:** Clara K. Chow, Julie Redfern.

**Validation:** Sonali Munot, Janet Bray, Adrian Bauman, Clara K. Chow, Julie Redfern.

**Visualization:** Sonali Munot, Janet Bray, Adrian Bauman, Simone Marschner, Clara K. Chow, Julie Redfern.

**Writing – original draft:** Sonali Munot.

**Writing – review & editing:** Sonali Munot, Janet Bray, Adrian Bauman, Emily J. Rugel, Leticia Bezerra Giordan, Simone Marschner, Clara K. Chow, Julie Redfern.

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
