## [Decision Letter · Decision Letter 0]

23 Jun 2022

PONE-D-22-15671Development of an intervention to facilitate dissemination of community-based training to respond to out-of-hospital cardiac arrest: FirstCPRPLOS ONE

Dear Dr. Munot,

Thank you for submitting your manuscript to PLOS ONE. After careful consideration, we feel that it has merit but does not fully meet PLOS ONE’s publication criteria as it currently stands. Therefore, we invite you to submit a revised version of the manuscript that addresses the points raised during the review process.

We look forward to receiving your revised manuscript.

Kind regards,

Dylan A Mordaunt, MD, MPH, FRACP

Academic Editor

PLOS ONE

Journal Requirements:

3. We noted in your submission details that a portion of your manuscript may have been presented or published elsewhere. 

"Yes, a paragraph that summarizes a section of the results has been mentioned in a related publication - a protocol paper for the FirstCPR study. A copy of the related work has been uploaded as a 'Related Manuscript file'."

4. We note that Supplementary information S3 includes an image of a [patient / participant / in the study]. 

Additional Editor Comments:

Thank you for your submission. The study presents the protocol development for a challenging intervention. Acknowledging our first reviewer's comments, protocols are a format PLoS One accepts.

With regards to the criteria for publication:

1. The study presents the results of original research.

2. Results reported have not been published elsewhere.

3. Experiments, statistics, and other analyses are performed to a high technical standard and are described in sufficient detail. The manuscript is a protocol, but also describes protocol development.

4. Conclusions are presented in an appropriate fashion and are supported by the data.

5. The article is presented in an intelligible fashion and is written in standard English.

6. The research meets all applicable standards for the ethics of experimentation and research integrity.

7. The article adheres to appropriate reporting guidelines and community standards for data availability. However, it would be worth the authors explicitly referencing a protocol checklist or guideline such as PRISMA-P or SPIRIT (or any relevant extensions), completing the checklist and submitting as an attachment in their resubmission.

Reviewers' comments:

Reviewer's Responses to Questions

**Comments to the Author**

1. Is the manuscript technically sound, and do the data support the conclusions?

Reviewer #1: No

Reviewer #2: Yes

2. Has the statistical analysis been performed appropriately and rigorously? 

Reviewer #1: N/A

Reviewer #2: N/A

3. Have the authors made all data underlying the findings in their manuscript fully available?

Reviewer #1: Yes

Reviewer #2: Yes

4. Is the manuscript presented in an intelligible fashion and written in standard English?

Reviewer #1: Yes

Reviewer #2: Yes

5. Review Comments to the Author

Reviewer #1: This submission poises a great problem to me. Ir is a lengthy description of building a tool, without ever examining its application. In short, this is a submission of "Introduction $ Materials and Methods" with no results.

I'd wait for results of applying this method.

Reviewer #2: Strengths:

This project meets a pressing community and public health need. The manuscript is written in clear language. This work is well positioned in the context of other outreach efforts. The theory-guided and collaborative approach involving multiple stakeholders is well reasoned and admirable. The key messages, encouraging hands-only CPR, Call, Push, Shock, etc. are on target.

Limitations:

One limitation is that this study does not assess the impact of the outreach program, which presumably will happen in the future.

The major criticism of the paper regards generalizability. This project appears to meet a local need in NSW Australia. What should a reader in the US or Europe or elsewhere take away from this? Why should this work matter to a global audience? Readers have no way of knowing whether the tool that was developed will work even in the location it is designed for, even less about how it might be useful more broadly.

Relating to that point, the authors mention that messaging needed to be matched to the specific audience and that resources with British and American accents and regional language such as “being sued” that might impair the impact of that messaging. Do you suggest that other places should similarly aim for messages that resonate with that specific area, using regional terms, referring to locally relevant shared cultural references, and avoiding accents?

Additional points:

Line 345. regarding “minimal adaptations”, do you mean in when generalized to other places in Australia? What about other areas? Are you suggesting that the tool you are developed will be broadly useful? or that the technique that you employed to bring the tool about will be broadly useful? Is specific tailoring a lesson that can be applied when outreach efforts are pursued elsewhere?

p 396. Was the survey group non-random in a way that could bias the results? On p 381. you write “Our approach also aims to increase the availability of education and training within communities that are often left out of such interventions.” Was the survey group (n=64) representative of the target audience and its diversity?

P 15. Avoid redundancy: There’s no point in providing a summary if it is almost exactly repeated in the response in quotes. Overall, Table 1 is okay, but I am not convinced of the need to include individual quotations unless there was a theme that emerged or if the response resonated with the investigators for some reason that should also be explained to the reader. It might be overstating a result to mention a survey response in the text (.e.g about jokes/comedy, line 275) and also in a summary statement followed by a respondent quotation saying the same thing. Was this one thing that one person said, or was it a theme that came up repeatedly?

P19. line 318. “slides”? I know what you mean of course, but younger I know what you mean, of course, readers might not. How about visual resource, or something like that?

6. PLOS authors have the option to publish the peer review history of their article (what does this mean?). If published, this will include your full peer review and any attached files.

Reviewer #1: **Yes: **Izhar Ben shlomo

Reviewer #2: No

---

## [Author Response · Author response to Decision Letter 0]

26 Jul 2022

We would like to thank PLOS ONE editors and reviewers for the time spent reviewing our manuscript and for the opportunity to respond to these useful comments. Responding to this feedback has strengthened the manuscript, as detailed below and via tracked changes in the manuscript. 

RESPONSE TO REVIEWERS’ COMMENTS

Strengths:

This project meets a pressing community and public health need. The manuscript is written in clear language. This work is well positioned in the context of other outreach efforts. The theory-guided and collaborative approach involving multiple stakeholders is well reasoned and admirable. The key messages, encouraging hands-only CPR, Call, Push, Shock, etc. are on target.

Limitations: 

One limitation is that this study does not assess the impact of the outreach program, which presumably will happen in the future. 

Yes, this is correct. This is a description of the intervention and its development. The evaluation and impact will be reported separately.

The major criticism of the paper regards generalizability. This project appears to meet a local need in NSW Australia. What should a reader in the US or Europe or elsewhere take away from this? Why should this work matter to a global audience? Readers have no way of knowing whether the tool that was developed will work even in the location it is designed for, even less about how it might be useful more broadly.

The takeaway message for a global audience should be that broad-brush programs may not be applicable in all contexts and settings and there is a need for culturally and locally appropriate programs that maximize collaborations and integrate locally produced materials. The overall process could provide a template for other international groups to design similar programs and localise them using materials specific to their culture, language, and accent. We have revised the manuscript to reflect this message in the following sections:

pg 2, lines 34-35; 

pg 3, lines 52 and 58;

pg 5, lines 88-90 and 99-100;

pg 6, line 101, 111;

pg 8, line 159;

pg 12, lines 230-234;

pg 22, lines 331, 345-347

pg 24, lines 381-387

Relating to that point, the authors mention that messaging needed to be matched to the specific audience and that resources with British and American accents and regional language such as “being sued” that might impair the impact of that messaging. Do you suggest that other places should similarly aim for messages that resonate with that specific area, using regional terms, referring to locally relevant shared cultural references, and avoiding accents? 

As described above, we believe it is important to localise materials for the target audience. As described in our paper, this includes incorporating shared cultural references, familiar accents and imagery that reflects local communities, so that it resonates as culturally familiar and appropriate. See amendments on pg 24 lines 381-387

Additional points:

Line 345. regarding “minimal adaptations”, do you mean in when generalized to other places in Australia? What about other areas? Are you suggesting that the tool you are developed will be broadly useful? or that the technique that you employed to bring the tool about will be broadly useful? Is specific tailoring a lesson that can be applied when outreach efforts are pursued elsewhere?

Yes, one of the key messages is that specific tailoring to the local context should be an important consideration along with seeking input from key stakeholders and using locally produced/created material. We have clarified this on pg 22-23, lines 378-384.

p 396. Was the survey group non-random in a way that could bias the results? On p 381. you write “Our approach also aims to increase the availability of education and training within communities that are often left out of such interventions.” Was the survey group (n=64) representative of the target audience and its diversity?

While the respondents were a non-random sample, we did maximise diversity of respondents from different age groups and socio-demographic backgrounds, including professionals in health and research as well as those from non-health occupations. We acknowledge that a representative sampling approach would be less biased and have noted this as a limitation:

pg 27, line 439, 447, and 448-453.

P 15. Avoid redundancy: There’s no point in providing a summary if it is almost exactly repeated in the response in quotes. Overall, Table 1 is okay, but I am not convinced of the need to include individual quotations unless there was a theme that emerged or if the response resonated with the investigators for some reason that should also be explained to the reader. It might be overstating a result to mention a survey response in the text (.e.g about jokes/comedy, line 275) and also in a summary statement followed by a respondent quotation saying the same thing. Was this one thing that one person said, or was it a theme that came up repeatedly?

We have now removed quotes in the updated Table 1 (pg 16) to just provide a summary of the preferences for material based on review and feedback from participants.

P19. line 318. “slides”? I know what you mean of course, but younger I know what you mean, of course, readers might not. How about visual resource, or something like that?

We agree with this point and have now reworded in several places:

pg 10, lines 189-190;

pg 11, line 211;

pg 22, lines 338-339; and

pg 23, line 352-355.

RESPONSE TO EDITORS’ COMMENTS 

Journal Requirements:

We have aligned the manuscript to meet PLOS ONE’s formatting and style requirements.

Phase two of this study involved sending out a link with shortlisted intervention material for participants to review, comment, and provide anonymous feedback. The link was sent via e-mail and consent was ‘implied’ if participants proceeded to click the link to provide anonymous feedback or they could choose to opt out by not proceeding to click the link. We have updated the manuscript on pg 10, lines 195-197 to reflect this implied consent.

3. We noted in your submission details that a portion of your manuscript may have been presented or published elsewhere. 

Yes, a paragraph that summarizes a section of the results was included in a protocol paper for the FirstCPR study. A copy of the related work has been uploaded as a 'Related Manuscript file'.

Yes, confirming that the related publication was peer-reviewed, formally published, and can be found at doi:10.1136/bmjopen-2021-057175 and cited as Munot S, Redfern J, Bray JE, et al. Improving community-based first response to out of hospital cardiac arrest (FirstCPR): protocol for a cluster randomised controlled trial. BMJ Open 2022;12:e057175. 

This publication describes the detailed protocol of the FirstCPR cluster randomised trial. Although it mentions the FirstCPR intervention, it does not detail the process of developing the intervention package, which is what the current manuscript aims to describe. We believe there is merit to describing this formative process in detail to provide generalizable knowledge on building a robust and scalable intervention package that can serve as a template for the development of similar large-scale public health and educational interventions around the globe. It was not possible to detail this process in the FirstCPR protocol publication and therefore merits a separate publication.

4. We note that Supplementary information S3 includes an image of a [patient / participant / in the study]. 

We appreciate the comment and have replaced these images with ones that do not contain any identifying information. 

The reference list has been checked and is complete and correct. At the time of submission Ref no.15 was in press and has since been published. This publication status has been updated accordingly on pg 31 line 537

Additional Editor Comments:

Thank you for your submission. The study presents the protocol development for a challenging intervention. Acknowledging our first reviewer's comments, protocols are a format PLoS One accepts.

With regards to the criteria for publication:

1. The study presents the results of original research.

2. Results reported have not been published elsewhere.

3. Experiments, statistics, and other analyses are performed to a high technical standard and are described in sufficient detail. The manuscript is a protocol, but also describes protocol development.

4. Conclusions are presented in an appropriate fashion and are supported by the data.

5. The article is presented in an intelligible fashion and is written in standard English.

6. The research meets all applicable standards for the ethics of experimentation and research integrity.

7. The article adheres to appropriate reporting guidelines and community standards for data availability. However, it would be worth the authors explicitly referencing a protocol checklist or guideline such as PRISMA-P or SPIRIT (or any relevant extensions), completing the checklist and submitting as an attachment in their resubmission.

All of these criteria have been met. 

With respect to criteria 5, we have made minor changes and fixed any grammar and tense where necessary to make it cleared to the reader. All changes are tracked. 

With respect to item 7, we were unable to find any existing checklists relevant to a protocol development arising from a co-design process; however, should you have a recommendation for a specific checklist that aligns with this format, we will be more than happy to complete and submit one.

Kind regards

Sonali Munot on behalf of all Authors

---

## [Editor Report · Decision Letter 1]

2 Aug 2022

Development of an intervention to facilitate dissemination of community-based training to respond to out-of-hospital cardiac arrest: FirstCPR

PONE-D-22-15671R1

Dear Dr. Munot,

We’re pleased to inform you that your manuscript has been judged scientifically suitable for publication and will be formally accepted for publication once it meets all outstanding technical requirements.

Kind regards,

Dylan A Mordaunt, MD, MPH, FRACP

Academic Editor

PLOS ONE

Additional Editor Comments (optional):

Thank you for your resubmission. This now meets the criteria for publication.
---

## [Editor Report · Acceptance letter]

15 Aug 2022

PONE-D-22-15671R1 

Development of an intervention to facilitate dissemination of community-based training to respond to out-of-hospital cardiac arrest: FirstCPR 

Dear Dr. Munot:

I'm pleased to inform you that your manuscript has been deemed suitable for publication in PLOS ONE. Congratulations! Your manuscript is now with our production department. 

Kind regards, 

on behalf of

Associate Professor Dylan A Mordaunt 

Academic Editor

PLOS ONE